# Multi-Mycotoxin Method Development Using Ultra-High Liquid Chromatography with Orbitrap High-Resolution Mass Spectrometry Detection in Breakfast Cereals from the Campania Region, Italy

**DOI:** 10.3390/toxins15020148

**Published:** 2023-02-12

**Authors:** Alfonso Narváez, Luana Izzo, Luigi Castaldo, Sonia Lombardi, Yelko Rodríguez-Carrasco, Alberto Ritieni

**Affiliations:** 1Department of Pharmacy, University of Naples Federico II, Via Domenico Montesano 49, 80131 Naples, Italy; 2Laboratory of Food Chemistry and Toxicology, Faculty of Pharmacy, University of Valencia, Av. Vicent Andrés Estellés s/n, Burjassot, 46100 Valencia, Spain; 3Health Education and Sustainable Development, Federico II University, 80131 Naples, Italy

**Keywords:** food safety, breakfast cereals, high-resolution mass spectrometry, mycotoxins

## Abstract

Breakfast cereals have been reported as one of the most susceptible cereal-based products to mycotoxin contamination. These products pose an even more concerning risk to human health since they are marketed as a ready-to-eat product and one of its main population targets is children. Therefore, the main goal of the present study was to conduct a monitoring study of multiple mycotoxins contained in breakfast cereals samples marketed in Italy through ultra-high performance liquid chromatography coupled to high-resolution Q-Orbitrap tandem mass spectrometry. An acetonitrile-based methodology was validated for quantifying 24 mycotoxins in breakfast cereals. The results showed that 93% of the samples contained at least one mycotoxin. Beauvericin was the most prevalent toxin (86% of samples; mean concentration: 30.66 µg/kg), although the main enniatins, zearalenone-derived forms and fumonisins B1 and B2 were also detected. Co-occurrence was observed in 73% of the positive samples with up to five mycotoxins simultaneously occurring, mainly due to the combination of beauvericin and enniatins. These results provided more evidence about the high impact of non-regulated mycotoxins, such as the emerging *Fusarium* toxins, in breakfast cereals, and encourages the development of analytical methodologies including these and zearalenone-derived forms that could be going unnoticed with current methodologies.

## 1. Introduction

Cereals and cereal-based products stand as one of the essential components of diets worldwide, regardless of the specific dietary habits related to different cultures [1]. The latest State of Food Security and Nutrition in the World (SOFI) released by the Food and Agriculture Organization of the United Nations (FAO) considered cereals as a staple food, accounting for about a third of the total diet [2]. Cereals provide a considerable amount of relevant nutrients, such as proteins, carbohydrates, vitamins, essential fatty acids and trace minerals [3]. Nevertheless, the chemical composition of cereals also offers a favorable environment for fungal contamination that can be further promoted by warm temperatures, high humidity or bad agronomic practices, among other factors [4,5]. Consequently, the appearance of fungi can lead to mycotoxin accumulation in the contaminated cereals.

Mycotoxins are toxic secondary metabolites produced by several fungi genera, including *Fusarium*, *Aspergillus*, *Penicillium*, *Claviceps* and *Alternaria* species. These mycotoxins can, therefore, accumulate within the crop, withstand the processing of raw materials and stay present in the final product of the food industry [6]. This could pose a health risk for consumers due to their wide variety of toxic effects, especially those derived from chronic dietary exposure, including immunotoxicity, reprotoxicity, nephrotoxicity or even carcinogenicity [7].

In order to reduce the impact of mycotoxins within the public health domain, authorities have established a regulatory framework in cooperation with scientific committees. One of the most relevant tools is represented by the maximum limits (MLs), released under Commission Decision 1881/2006 and its amendments, which established the maximum concentration permitted for a mycotoxin/group of mycotoxins within a specific food product, according to several variables, such as its susceptibility to accumulation, degree of processing, population target, etc [8]. Therefore, susceptible crops and final products are in constant surveillance in order to guarantee safe consumption [9].

Among cereal-based products, breakfast cereals have been reported as one of the most susceptible cereal-based products to mycotoxin accumulation worldwide [10,11]. In addition, this type of product poses an even more concerning risk to human health due to two reasons: it is marketed as a ready-to-eat product, meaning that it does not undergo any processing that could lower the total burden of mycotoxins [12] and, lastly, one of its main relevant population targets is children and teenagers [13]. Taking all of these into consideration, MLs have been set for deoxynivalenol (DON) at 500 µg/kg, zearalenone (ZEN) at 50, µg/kg the sum of fumonisins B1 and B2 (FB1 and FB2) at 400 µg/kg, aflatoxin B1 (AFB1) at 2 µg/kg and the sum of aflatoxins B1, B2, G1 and G2 (AFB2, AFG1, AFG2) at 4 µg/kg. Nevertheless, other mycotoxins, such as enniatins (ENNs) that are not regulated in breakfast cereals, could also be occurring, so human exposure to mycotoxins through breakfast cereal consumption may be underestimated. In this context, the monitoring of these toxins is still encouraged by the Panel on Contaminants in the Food Chain from the European Food and Safety Authority (EFSA CONTAM Panel) [14].

The analysis of mycotoxins contained in breakfast cereals has been carried out through liquid chromatography coupled to tandem mass spectrometry (LC-MS/MS), which is considered the gold-standard analytical technique for food contaminants [15]. Nevertheless, sample preparation can be very variable across studies depending on the targeted mycotoxins, availability of samples, etc. This is reflected by several recent methodologies for the quantification of mycotoxins in breakfast cereals that use different approaches, such as enzyme-linked immunoassays (ELISA) procedures or solid-liquid extraction for specific groups of mycotoxins. Nevertheless, cheap, simple, universal and less-time-consuming procedures are preferred for the simultaneous extractions of chemically different mycotoxins. Therefore, the main goal of the present study was to conduct a monitoring study of multiple mycotoxins (*n* = 24) contained in 14 typologies of breakfast cereals marketed in Italy through ultra-high-performance liquid chromatography coupled to high-resolution Q-Orbitrap tandem mass spectrometry (UHPLC-Q-Orbitrap-HRMS). To achieve this, an acetonitrile-based procedure was optimized and validated alongside the analytical methodology.

## 2. Results and Discussion

### 2.1. Optimization of Sample Preparation

Among all the different methodologies that have been developed for the extraction of mycotoxins from cereals and extensively reviewed in the literature [16], three procedures were selected according to the typology of matrix and the targeted mycotoxins, and later compared in terms of recoveries and matrix effect. Targeted mycotoxins were: aflatoxins B1, B2, G1 and G2 (AFB1, AFB2, AFG1 and AFG2), beauvericin (BEA), DON, enniatins A, A1, B and B1 (ENNA, ENNA1, ENNB and ENNB1), FB1, FB2, fusarenon X (FUS-X), HT-2 toxin (HT-2), neosolaniol (NEO), ochratoxin A (OTA), T-2 toxin (T-2), α-zearalenol (α-ZEL), α-zearalanol (α-ZAL), β-zearalenol (β-ZEL), β-zearalanol (β-ZAL), zearalanone (ZAN) and ZEN. All the selected extraction methodologies were acetonitrile-based and their features are shown in Table 1.

To compare the proposed methodologies, spiking experiments were carried out at 50 µg/kg and calibration curves both in neat solvent and a blank matrix were built within a concentration range from 200 to 0.2 ng/mL. The results are shown in Figure 1.

Method A was able to recover a high percentage (79%) of the total number of mycotoxins (*n* = 24) within a suitable range of recovery (70–120%), whereas the 21% of mycotoxins left did not reach the minimum value required for a proper validation. Method B was the least effective procedure for recovering mycotoxins from breakfast cereals, with only 21% of the total mycotoxins compiling the established range whereas the rest of mycotoxins were poorly recovered. Finally, Method C showed the best performance with 100% of the mycotoxins recovered within the range of recovery of 70–120%.

The quantity of solvent used in the extraction step seems to be critical for recovering mycotoxins according to the observed results. In this line, the most efficient methodology applied the lowest volume of acetonitrile, meaning that using too much solvent of extraction may lead to an excessive dilution of the analytes hampering their further determination through LC-HRMS. This ratio sample:solvent of 1:4 (*w/v*) has been previously reported as optimal by other multi-mycotoxin studies using cereals as a matrix [17].

In terms of the matrix effect, the signal suppression/enhancement (SSE) effect was used to compare the three studied methodologies. Method A showed a high percentage (79%) of mycotoxins with negligible interference (80 < %SSE < 120) whereas the rest of the mycotoxins showed a suppression effect (%SSE < 80%). Method B, again, showed the worst performance with 63% of the total mycotoxins presenting signal suppression and only 37% of mycotoxins fitting the required range. Method C reflected negligible interference from the matrix for all the assayed mycotoxins, probably due to a dilution in a high quantity of water that seemed to be able to remove matrix interferents prior to the extraction step with acetonitrile. These results showed that this typology of samples contained interferents, forcing us to introduce a step to remove them prior to analysis. Method B and Method C used both water as a way to minimize the interference of the matrix by dilution. In light of the obtained results, the amount of water used in Method B was not enough to reach a suitable SSE. Method A, instead, used a clean-up step for the removal of contaminants using C18 that was also shown to be insufficient to achieve a negligible matrix effect. Therefore, due to its better performance, Method C was further subjected to validation study.

### 2.2. Method Validation

The proposed methodology (Method C) was later validated according to Commission Decision 2002/657/EC [18] in terms of trueness, precision, linearity, limits of quantification (LOQ) and selectivity. The results are shown in Table 2. Calibration curves built in a blank matrix and neat solvent reflected coefficients of determination (R^2^) above 0.990 in the evaluated linear range (200 ng/mL to LOQ), confirming a deviation below 20% considering the theoretical value of each calibration point. The matrix effect was evaluated throughout the %SSE by comparing the slopes of both curves, showing negligible interference for all the assayed mycotoxins (80–120%). A proper selectivity was also ensured by analyzing blank samples (*n* = 20) and observing no co-eluants peaks within the retention times attached to each mycotoxin. High sensitivity was displayed by the present methodology, allowing us to quantify all analytes at limits within a range 0.20–12.5 µg/kg, enough to check whether the samples complied with the current legislation: DON at 500 µg/kg, ZEN at 50 µg/kg, the sum of FB1 and FB2 at 400 µg/kg, AFB1 at 2 µg/kg and the sum of AFB1, AFB2, AFG1 and AFG2 at 4 µg/kg. Recovery studies were carried out throughout spiking experiments at three fortification levels (50, 25 and 10 µg/kg) during three non-consecutive days. Recoveries were shown to be suitable according to legislation (70–120%) with intra-day and inter-day precision values below 20% [18]. This in-house validated methodology for monitoring up to 24 mycotoxins was further applied in marketed breakfast cereal samples, ensuring the reliability of the obtained results.

The more recent methodologies for the multi-mycotoxin analysis of breakfast cereals are based on the use of different sample treatments. In this line, several enzymatic assays have been used by Foerster et al. [19] for the quantification of total AFs, OTA, total FBs, ZEN, DON, HT-2 and T-2. Similarly, Mallmann et al. [20] designed or adapted different protocols based on solid-liquid extraction for the quantification of mycotoxins, grouped as follows: total AFs; ZEN and DON; FB1 and FB2; T-2, HT-2, FUS-X and other trichothecenes; OTA. The here-validated methodology allowed us to unify all the above-mentioned mycotoxins into one single sample treatment through a simple acetonitrile-based extraction that provided high sensitivity, an easy workflow and stood as a less time-consuming alternative compared to the most recent strategies applied in breakfast cereal samples.

### 2.3. Analysis of Real Samples

Once validated, the methodology was applied to commercially available breakfast cereal samples. Mycotoxins found in the samples and their occurrence data are shown in Table 3.

Altogether 93% of the samples contained at least one mycotoxin, highlighting the frequent contamination to which these products are subjected. The most prevalent mycotoxin was BEA, present in 86% of the samples at a mean concentration of 6.70 µg/kg, whereas other emerging *Fusarium* toxins such as ENNs were also detected but with less frequency (7–21%). At a quantitative level, ENNB and B1 were the most relevant ENNs, occurring at mean concentrations of 23.09 and 15.03 µg/kg, respectively. ZEN-derived forms α- and β-ZEL were also present in 14 and 7% of the samples, respectively, at levels below 6 µg/kg. In addition, the regulated mycotoxins FB1 and FB2 were also identified in the assayed samples. FB1 was relatively frequent, occurring in 36% of the samples at a mean concentration of 22.18 µg/kg, whereas FB2 showed less relevance being quantified in 14% of samples at a low mean concentration level of 1.61 µg/kg. Currently, ML for FBs in breakfast cereals are established at 400 µg/kg for the combination of both toxins, so all the here-analyzed samples complied with the legislation. Unlike other studies on breakfast cereal samples, none of the samples tested positive for DON, which is known for its impact in wheat and maize-based products. Mitigation strategies have been extensively discussed over the last years in order to reduce the impact of DON and other relevant mycotoxins in foods and feeds, especially pushed forward by initiatives such as the MycoKey project, that provided practical guides for achieving this reduction on the burden of those mycotoxins [21]. In this line, a meta-analysis of DON and other trichothecenes in cereal-based products revealed a decrease over the years regarding concentration and prevalence [10,11].

Out of all positive samples, co-occurrence was observed in 73% of the samples. This event was mainly due to the combination of BEA with other ENNs (46% of samples with co-occurrence), FB1 (27%) and α-ZEL (27%). Up to five mycotoxins co-occurred within the same sample.

The analysis of multiple mycotoxins in cereals marketed in Italy is a well-studied topic, but the majority of the previous studies focused on grains, meaning the raw materials, whereas final products such as breakfast cereals have been scarcely studied. In this line, Capei et al. [22] evaluated the presence of OTA in breakfast cereals marketed in Italy, finding contamination in 8% of the samples (*n* = 25) at concentrations below 2 µg/kg. The low impact of OTA in breakfast cereals can relate to the here-studied samples, that were not contaminated with OTA. This could indicate that, regardless the potential contamination within the raw material, the mitigation strategies or the type of processing related to this kind of products could affect the total burden of OTA. Nevertheless, the presence of *Fusarium* toxins has been more frequently reported in these cereal-based products rather than *Aspergillus*. Lo Magro et al. [23] analyzed the presence of FB1 and FB2 in breakfast cereals (*n* = 35), reporting 28% of positive samples for, at least, one of both mycotoxins within a concentration range of 35–178 µg/kg (mean concentration 89 µg/kg). In terms of prevalence, these results were similar to those here-presented, whereas at a quantitative level, mean concentration values resulted to be almost twice the values reported in the present study. It has to be taken into consideration that the sensitivity of analytical techniques has greatly improved over the last years; whereas LOQs reported by Lo Magro et al. [23] were 13 and 16 µg/kg for FB1 and FB2, respectively, this study provided much higher sensitivity with LOQs at 3.12 and 0.39 µg/kg. The differences in mean concentrations could be then due to the differences regarding the sensitivity. The presence of other *Fusarium* mycotoxins, such as DON, ZEN, T-2 and HT-2, in breakfast cereal marketed in Italy (*n* = 43) was also studied by Romagnoli et al. [24], remarking a considerable impact of DON. This mycotoxin was identified in 37% of the samples, followed by ZEN (9%), T-2 and HT-2 (5%). In the here-presented results, neither DON, T-2 nor HT-2 were detected in any of the assayed samples, whereas the ZEN-derived forms α-ZEL and β-ZEL were detected in a similar proportion to ZEN when compared to that previous study. Therefore, considering that the EFSA CONTAM Panel established a tolerable daily intake (TDI) for ZEN in combination with its derived-forms through a relative potency factor system [25], these mycotoxins should also be included in multi-mycotoxin methodologies involving this matrix of study.

Of special interest is the great impact of emerging *Fusarium* toxins in the here-analyzed samples. Evidence about the presence of ENNs and BEA in breakfast cereals has been long reported until now, not only in cereal-based products marketed in Italy but also worldwide [26,27,28,29,30]. Therefore, in this context, the present study provides more evidence about the ubiquity of these toxins that are not regulated and whose toxicology remains under investigation. Although other surveys have assessed the presence of mycotoxins in breakfast cereals marketed in other countries [19,20,31], ENNs and BEA are not usually included in the analytical methodologies and the focus is mainly placed on DON, ZEN, AFs and FBs, which are the regulated toxins. Furthermore, extensive exposure to ENNB and ENNB1 has been observed by previous biomonitoring studies in the Italian population [32,33], encouraging the monitoring of these toxins in food products in order to have a more in-depth knowledge about their impact as recommended by the EFSA CONTAM Panel [14].

## 3. Conclusions

A simple acetonitrile-based methodology was optimized and validated for the analysis of 24 mycotoxins in breakfast cereal samples. Then, it was applied to samples marketed in Italy (*n* = 14) belonging to different commercial brands.

Emerging *Fusarium* toxins showed the highest relevance, especially BEA, which was identified in 86% of the samples and co-occurring with other ENNs in most of the cases. FBs were also detected in a few samples but complying with the current legislation, whereas ZEN-derived forms α-ZEL and β-ZEL were detected too.

The here-presented results provided more evidence about the extensive presence of non-regulated mycotoxins in breakfast cereal samples, which are not usually assessed when monitoring such matrices. Despite the lack of legislation concerning these emerging *Fusarium* toxins, analytical methodologies must include them considering their ubiquitous character and the heavy exposure reported after human biomonitoring studies in the Italian population. Additionally, the high rate of co-occurrence found in the here-analyzed samples could represent a general concern due to the toxicity developed by mixtures of mycotoxins; even more when structurally-similar mycotoxins, such as emerging *Fusarium* toxins, have been reported to display additive and synergistic effects. Therefore, this study encourages a more in-depth study about the occurrence of not only regulated, but non-regulated mycotoxins in cereal-based products that could provide a better overview about the impact of those mycotoxins in food and, by extension, in human health.

## 4. Material and Methods

### 4.1. Chemical, Reagents and Materials

Water and methanol (MeOH) for LC mobile phase (LC-MS grade) and acetonitrile (ACN) were provided by Merck (Darmstadt, Germany). Ammonium formate (analytical grade) was supplied by Fluka (Milan, Italy). Formic acid (MS grade) was acquired from Carlo Erba reagents (Cornaredo, Italy). Sodium chloride (NaCl) and octadecyl carbon chain-bonded silica (C18) (analytical grade) and were provided by Sigma Aldrich (Milan, Italy). Conical centrifuge polypropylene tubes of 50 and 15 mL were provided by BD Falcon (Milan, Italy). Syringe filters with polytetrafluoroethylene membrane (PTFE, 15 mm, diameter 0.2 µm) were supplied by Phenomenex (Castel Maggiore, Italy).

Analytical standards of the following mycotoxins (HPLC purity > 98%): AFB1, AFB2, AFG1, AFG2, BEA, dDON, ENNA, ENNA1, ENNB, ENNB1, FB1, FB2, FUS-X, HT-2, NEO, OTA, T-2, α-ZEL, α-ZAL, β-ZEL, β-ZAL, ZAN and ZEN were purchased from Sigma-Aldrich (Milan, Italy).

For each analytical standard, a stock solution was prepared by dissolving 1 mg in 1 mL of MeOH. Afterwards, working solutions were built by properly diluting in MeOH/H_2_O (70:30 *v/v*) 0.1% formic acid until reaching the desired concentrations for spiking experiments (50, 25 and 10 µg/kg). Working solutions were stored in securely closed vials at −20 °C.

### 4.2. Sampling

Fourteen different typologies of ready-to-eat breakfast cereals were included in the present study. The ready-to-eat products were based on (i) only corn (79% of samples); (ii) rice, wheat and barley (14% of samples); and (iii) wheat and oat (7% of samples). Three different confections of each product belonging to the same lot were acquired from local supermarkets located in Campania region, Southern Italy, during February and March 2022. The three confections from each typology of product were milled, homogenized and analyzed as a composite sample, in order to increase their representativity. Once homogenized, samples were kept in dark and cool conditions as recommended by the manufacturer. Analyses were carried out within five days after receiving them.

### 4.3. Sample Treatment

An acetonitrile-based methodology was applied for the extraction of mycotoxins from breakfast cereals samples. In brief, 2.5 g of sample was weighted into 50 mL conical centrifuge polypropylene tubes and 15 mL of distilled water was added. The samples were then vortexed (ZX3; VEPL Scientific, Usmate, Italy) for 2 min and 10 mL of ACN 0.1% formic acid was later added into the tubes. After vortexing for another 2 min, the samples were sonicated for 10 min (LBS 1; Zetalab srl, Padua, Italy). A salting-out step was performed using 2 g of NaCl followed by manual agitation. The samples were then refrigerated at −80 °C for 5 min and centrifuged at 4907× *g* for 10 min (SL16R, Thermo Fisher Scientific, Kalkberg, Germany). Then 4 mL of supernatant was transferred into 15 mL conical centrifuge polypropylene tubes and evaporated to dryness under gentle nitrogen flow at 40 °C. Lastly, extracts were resuspended in 500 µL of MeOH:H_2_O (70:30 *v/v*) 0.1% formic acid and ammonium formate 5 mM and filtered through a 0.22 μm filters prior to UHPLC-Q-Orbitrap HRMS analysis.

During the optimization stage, another two similar procedures were evaluated using the same amount of sample (2.5 g). Method B followed the same procedure as above but using 4 mL of distilled water and 16 mL of ACN 0.1% formic acid. On the other hand, to carry out Method A, 20 mL of ACN 1% formic acid was added to conical centrifuge polypropylene tubes containing 2.5 g of sample. The samples were vortexed for 2 min and sonicated for 10 min. Then they were refrigerated at −80 °C for 5 min and centrifuged at 4907× *g* for 10 min. Then 4 mL of the supernatant was transferred into 15 mL conical centrifuge polypropylene tubes that contained 200 mg C18 and 900 MgSO_4_. The tubes were vortexed for 1 min and centrifuged at 4907× *g* for 10 min. The supernatants were transferred into separate 15 mL conical centrifuge polypropylene tubes and evaporated to dryness under gentle nitrogen flow at 40 °C.

### 4.4. UHPLC-Q-Orbitrap HRMS Analysis

Detection and quantification of the targeted mycotoxins was performed through a UHPLC-Q-Orbitrap HRMS methodology previously optimized by Izzo et al. [34].

Chromatographic analysis was carried out through a Dionex UltiMate^®^ 3000 system consisting of a quaternary UHPLC pump working at 1250 bar (125 MPa), a degassing system, an autosampler device and a thermostatically controlled column (ThermoFisher Scientific, Waltham, MA, USA). Chromatographic separation of mycotoxins was performed using a thermostated Kinetex Biphenyl 2.6-μm column (100 × 2.1 mm, Phenomenex) (T = 25 °C) coupled to a guard column (5 × 2 mm, 1.8-μm particle size) and an inline filter kept at 30 °C. Water (A) and methanol (B), both containing 0.1% formic acid and 5 mM ammonium formate were used as mobile phases. The chromatographic linear gradient was set as follows: 0 to 0.5 min, 10% B; 0.5 to 2.5 min, up to 80% B; 2.5 to 5.5 min, 100% B; 5.5 to 7.5 min, down to 10% B. Lastly, the gradient went back to 10% B and was held for 1.5 min for column re-equilibration. The total run time was 9 min with an injection volume of 5 μL and a flow rate of 0.5 mL/min.

Mass spectrometry analysis was performed through a Q-Exactive Orbitrap, which operated in both positive and negative ion modes by setting 2 scan events: full scan (FS) and all ion fragmentation (AIF). The ion source parameters were set as follows: capillary temperature 310 °C, S-lens RF level 50, spray voltage 2.8 kV (-kV en ESI- mode), auxiliary gas heater temperature 305 °C, sheath gas pressure (N_2_ > 95%) 35 arbitrary units and auxiliary gas (N_2_ > 95%) 10 arbitrary units. The parameters for FS mode were: resolving power of 35,000 full width half maximum (FWHM, defined for *m/z* 200), automatic gain control (AGC) target 1 × 10^6^, scan range 90–1000 *m/z*, injection time set to 200 ms and scan rate set at 3 scans/s, whereas the AIF mode used the following settings: scan rate 3 scans/s, scan time 0.10 s, maximum injection time 200 ms, mass resolving power 17,500 FWHM, ACG target 1 × 10^5^, scan range 80–1000 *m/z*, retention time window 30 s and isolation window 5.0 *m/z*. Collision energies set for AIF event ranged from 10 to 60 eV, based on those previously optimized for each analyte in a previous study [30]. Alongside precursor ions, two product ions per analyte were monitored with a maximum mass error of 5 ppm for ensuring a proper identification according to legislation. The mass spectrometer was regularly calibrated using calibration solutions provided by ThermoFisher during three-day intervals and before each sequence. Retention time, elemental composition, theoretical and measured mass, accurate mass error, collision energy and product ions for the analyzed compounds are shown in Table 4. Data treatment was carried out within the Xcalibur platform v.3.1.66.10.

### 4.5. Method Validation

An in-house validation study was conducted according to the EU Commission Decision 2002/657/EC guidelines referring to linearity, selectivity, trueness and sensitivity expressed as LOQs [18]. Linearity was determined by injecting a series of neat solvent and matrix-matched calibration curves at ten concentration levels ranging from 200 to 0.2 ng/mL, accepting a maximum deviation of 20% for each calibration level. The coefficient of determination was calculated using the means of the least square approach. The slopes corresponding to the calibration curve built in neat solvent (A) and blank matrix (B) were compared in order to elucidate any SSE effect. Therefore, the ratio (B/A × 100) may reflect a suppression (ratio < 100%) or enhancement (ratio > 100%) that could influence the quantitative results. The selectivity of the method was assessed by injecting blank samples (*n* = 20), observing no peaks coeluted within the same retention time area as the analytes, always considering a mass error of 5 ppm. Recovery studies were conducted by spiking three blank samples at three different fortification levels: 50, 25 and 10 µg/kg. The measurements were made during three non-consecutive days. Values ranging from 70 to 120% of recovery were considered as optimal. Precision was assessed in terms of repeatability (relative standard deviation after three determinations in a single day, RSDr) and reproducibility (relative standard deviation after determinations in triplicate on three non-consecutive days, RSDR). Sensitivity was determined through the LOQ for each analyte as follows: after evaluating the linear range, LOQs were determined as the lowest calibration level inside the linear range, considering a maximum deviation of 20% compared to the theoretical value and a mass error below 5 ppm for each analyte.

### 4.6. Quality Control/Quality Assurance

Spectral and chromatographic data were combined for the correct identification of the analytes. Retention times attached to the assayed analytes were compared in both positive samples and standards in neat solvent at a tolerance of ±2.5% of the total run time (9 min). Data quality was verified through the inclusion of a comprehensive range of quality assurance and quality control procedures. Each batch of samples included a procedural blank, a reagent blank and a matrix-matched calibration in order to evaluate the robustness and stability of the system throughout the whole analysis.

### 4.7. Statistical Analysis

Statistical data treatment was carried out in software package IBM SPSS v.25. Levene’s test was applied in order to evaluate the normality of data distribution. Kruskal–Wallis’ test was used for assessing significant differences among results. A confidence level of 95% was settled for data treatment and a *p*-value < 0.05 was considered as significant.

## Figures and Tables

**Figure 1 toxins-15-00148-f001:**
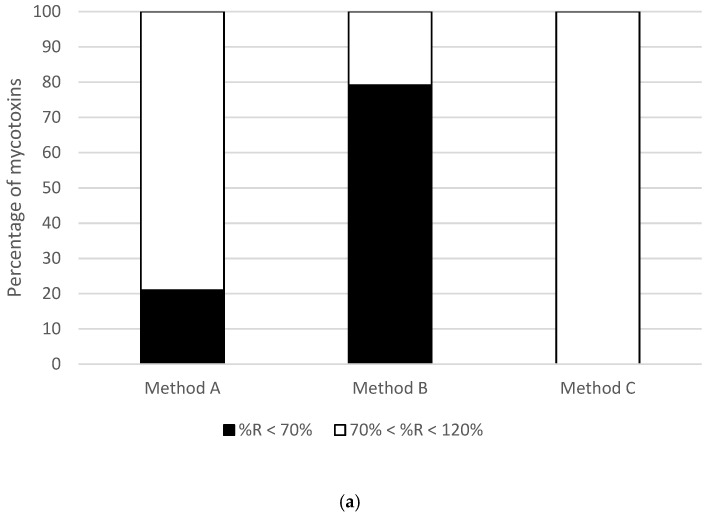
Percentage of mycotoxins (**a**) recovered ^®^ below the minimum required (<70%), inside the suitable range (70–120%) or above the range (>120%) for the assayed methodologies; (**b**) showing a signal suppression/enhancement effect (SSE) below the minimum required (<80%), inside the suitable range (80–120%) or above the range (>120%) for the assayed methodologies.

**Table 1 toxins-15-00148-t001:** Features of the three sample treatment methodologies compared.

Features	Method A	Method B	Method C
Amount of sample (g)	2.5	2.5	2.5
Dilution in water (mL)	-	4	15
Quantity of solvent (mL)	20	16	10
Acidification with formic acid (%)	1.0	0.1	0.1
Clean up	200 mg C18 + 900 MgSO_4_	-	-

**Table 2 toxins-15-00148-t002:** Method performance.

			Recovery (%)			Precision (%) [RSDr, (RSDR)]			
Analyte	Linearity (R2)	SSE (%)	50 µg/kg	25 µg/kg	10 µg/kg	50 µg/kg	25 µg/kg	10 µg/kg	LOQ (µg/kg)
DON	0.9969	107	107	115	117	12 (8)	13 (9)	13 (11)	12.5
NEO	0.9948	81	105	108	118	10 (9)	9 (14)	7 (7)	0.78
HT-2	0.9981	96	111	112	91	6 (8)	5 (4)	10 (13)	0.78
T2	0.9982	85	102	115	104	10 (11)	17 (11)	15 (11)	0.39
BEA	0.9986	85	106	120	100	14 (10)	9 (7)	14 (10)	0.20
ENNA	0.9978	82	107	120	97	16 (13)	5 (5)	11 (11)	0.20
AFB1	0.9940	83	104	107	76	14 (11)	10 (15)	15 (12)	1.56
ENNA1	0.9980	99	108	106	119	14 (10)	7 (6)	15 (10)	1.56
ENNB	0.9903	98	106	119	113	13 (10)	10 (10)	7 (9)	0.39
ENNB1	0.9903	99	107	112	116	6 (12)	6 (4)	12 (8)	1.56
AFG1	0.9924	96	100	109	97	14 (14)	7 (9)	11 (7)	0.78
AFG2	0.9921	95	113	115	119	6 (5)	13 (12)	14 (10)	0.78
AFB2	0.9960	90	101	119	119	17 (10)	12 (19)	5 (7)	0.20
OTA	0.9932	87	109	115	115	6 (7)	11 (7)	9 (11)	6.25
FUS X	0.9914	92	108	105	119	9 (9)	17 (11)	2 (4)	12.5
FB2	0.9962	82	105	98	112	6 (12)	12 (11)	13 (13)	0.39
FB1	0.9993	85	109	111	114	13 (17)	11 (23)	8 (9)	3.12
α-ZEL	0.9947	94	105	105	114	7 (8)	19 (16)	9 (5)	3.12
α-ZAL	0.9935	89	99	94	87	9 (11)	10 (7)	11 (14)	1.56
β-ZEL	0.9972	84	107	108	117	6 (6)	14 (11)	13 (10)	1.56
β-ZAL	0.9936	91	103	98	91	5 (8)	9 (10)	11 (13)	1.56
ZAN	0.9930	86	97	111	103	5 (3)	10 (11)	5 (6)	1.56
ZEN	0.9980	97	107	107	111	11 (10)	16 (11)	14 (16)	1.56

**Table 3 toxins-15-00148-t003:** Occurrence of mycotoxins in the here-analyzed breakfast cereal samples.

Mycotoxin	Prevalence (%)	Mean Concentration (µg/kg)	SD (µg/kg)	Maximum (µg/kg)
BEA	86	6.7	7.74	30.66
ENNA	7	1.05	-	1.05
ENNA1	21	6.34	7.35	14.79
ENNB	21	23.09	17.55	35.39
ENNB1	14	15.03	17.57	27.45
FB2	14	1.61	0.97	2.3
FB1	36	22.18	20.4	55.51
α-ZEL	14	5.79	0.23	5.96
β-ZEL	7	4.39	-	4.39

**Table 4 toxins-15-00148-t004:** UHPLC-Q-Orbitrap parameters for the assayed mycotoxins.

Analyte	Retention Time (min)	Elemental Composition	Adduct	Theoretical Mass (*m/z)*	Measured Mass (*m/z*)	Accuracy (Δ ppm)	Collision Energy (eV)	Product Ions (*m/z*)
			Ion					
DON	2.69	C_15_H_20_O_6_	[M+H]^+^	297.13326	297.13345	0.64	13	249.11194; 203.10648
FUS-X	3.58	C_17_H_22_O_8_	[M+Na]^+^	377.12073	377.12063	−0.27	20	228.16002; 175.07550
NEO	3.76	C_19_H_26_O_8_	[M+NH_4_]^+^	400.19659	400.19632	−0.67	10	305.13803; 141.0053
HT-2	4.3	C_22_H_32_O_8_	[M+NH_4_]^+^	442.24354	442.24323	−0.7	27	263.12744; 215.10641
β-ZEL	4.34	C_18_H_24_O_5_	[M-H]^−^	319.1551	319.155	−0.31	36	174.95604; 160.97665
α-ZEL	4.44	C_18_H_24_O_5_	[M-H]^−^	319.1551	319.155	−0.31	36	174.95604; 129.01947
T-2	4.49	C_24_H_34_O_9_	[M+NH_4_]^+^	484.25411	484.2543	0.39	23	215.10603; 185.09561
ZAN	4.68	C_18_H_24_O_5_	[M-H]^−^	319.1551	319.155	−0.31	35	273.01187; 131.05020
ZEN	4.7	C_18_H_22_O_5_	[M-H]^−^	317.13945	317.13928	−0.54	-32	175.03989; 131.05008
AFG1	4.73	C_17_H_12_O_7_	[M+H]^+^	329.06558	329.06549	−0.27	40	243.06467; 200.04640
AFG2	5.03	C_17_H_14_O_7_	[M+H]^+^	331.08123	331.08078	−1.36	37	313.07010; 245.08032
ENNB	5.15	C_33_H_57_N_3_O_9_	[M+NH_4_]^+^	657.44331	657.44348	0.26	50	214.1432; 196.1328
AFB2	5.16	C_17_H_14_O_6_	[M+H]^+^	315.08631	315.08615	−0.51	36	287.09064; 259.05945
ENNB1	5.18	C_34_H_59_N_3_O_9_	[M+NH_4_]^+^	671.45986	671.45935	−0.76	48	214.14343; 196.13295
ENNA1	5.24	C_35_H_61_N_3_O_9_	[M+NH_4_]^+^	685.47461	685.47449	−0.18	48	228.15900; 210.14847
AFB1	5.26	C_17_H_12_O_6_	[M+H]^+^	313.07066	313.07053	−0.42	36	285.07489; 269.04373
ENNA	5.28	C_36_H_63_N_3_O_9_	[M+NH_4_]^+^	699.49026	699.48987	−0.56	43	228.15900; 210.14847
BEA	5.4	C_45_H_57_N_3_O_9_	[M+NH_4_]^+^	801.44331	801.44339	0.1	35	262.76715; 244.18239
FB1	6.03	C_34_H_59_NO_15_	[M+H]^+^	722.39575	722.39539	−0.5	48	352.32010; 334.30963
OTA	6.5	C_20_H_18_NO_6_Cl	[M+H]^+^	404.08954	404.08931	0.57	16	358.08304; 341.05658
FB2	6.78	C_34_H_59_NO_14_	[M+H]^+^	706.40083	706.40192	−1.54	58	336.32547; 318.31488

## Data Availability

The data presented in this study are available in this article.

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
