# Peer review of "Multi-Mycotoxin Method Development Using Ultra-High Liquid Chromatography with Orbitrap High-Resolution Mass Spectrometry Detection in Breakfast Cereals from the Campania Region, Italy"

_toxins, 2023, doi:10.3390/toxins15020148_

Round 1

Reviewer 1 Report

The authors report the development of a multi-mycotoxin method for the analysis of several mycotoxins in breakfast cereals and results of the application of this method in a small study with 14 samples from the Italian market.

The method development and validation seems to be scientifically sound in general, but some details still have to be addressed. The results of the market study also need more clarification and discussion.

Taken together the manuscript should not be published in its present form and needs some major revisions before publication.

Specific remarks:

1) Picking up the phrase “multi-mycotoxin method” from the title the authors should explain the selection of mycotoxins they included in their method. Apart from the regulated mycotoxins, the choice of “emerging mycotoxins” seems rather random. Why was fusarenon X (FUS-X) included, but not the structural related nivalenol? Why did the authors omit the well-known deoxynivalenol (DON) derivatives 3- and 15-acetyl-DON and – most important – DON-3-glucoside? All these compounds are commercially available and have been detected in cereals before, so this point needs some clarification.

2) It is completely astonishing that the authors did not detect deoxynivalenol (DON) in any of their samples. Both literature and legislation are addressing DON as major concern in breakfast cereals, which is contradictory to the results of the authors.

However, the product group “breakfast cereals” is very general and includes different products that have been processed in varying ways that may alter mycotoxin contamination. Authors should give more details about their samples, because this might help to clarify the mycotoxin distribution.

Furthermore, it would have been helpful, if the authors had determined real trueness of the method (not recovery as detailed below) by the analysis of certified reference materials or participation in a ring test. This would have dispelled any concerns, if this non-finding of DON in the samples might be a methodical issue.

Anyway, the authors should address this topic in more detail in the manuscript and discuss potential reasons for their non-finding of certain mycotoxins in their samples.

3) A value named “coefficient of linearity (r2)” (line 122) is not existing. This phrase must not be used. The authors are referring to the “coefficient of determination (R2)”, which is a descriptive value obtained from linear regression. However, the coefficient of determination is a very poor criterion to assess linearity and should not be used anymore nowadays. There are better methods to address this topic like the Lack-of-fit-test or Mandel’s test. Please confer: https://www.researchgate.net/publication/46430384_On_the_misuse_of_the_correlation_coefficient_to_assess_linearity_of_calibration_curves

4) Trueness (line 327) cannot be determined by spiking experiments, but only by analyzing certified reference material (CRM). Commission Regulation 2002/657/EC allows that trueness of measurements is assessed through recovery studies, if no CRMs are available. However, there are CRMs available for some mycotoxins in cereals, so this exception from the rule cannot be claimed by authors in general. The validation studies that they have performed were recovery studies and should be named like that.

5) Table 2 and throughout the whole manuscript: It is common practice in the mycotoxin field to report quantitative values as “µg/kg” and not “ng/g”. Legal limits for mycotoxins in Commission Regulation 1881/2006 are also given in “µg/kg” and the authors are asked to adapt their values accordingly.

6) The authors should give more details how they determined their LOQ. Were the values obtained from spiking the blank matrix?

7) What kind of breakfast cereal was used as blank matrix for evaluation of the matrix effect?

8) Figure 1: please check if a) and b) need to be swapped to fit the description in the figure caption. The abbreviations “%R” and “%SSE” should also be defined in the figure caption to facilitate reading.

Author Response

Journal Toxins (ISSN 2072-6651)

Manuscript ID: toxins-2199068

Title: Multi-mycotoxin method development using ultra-high liquid chromatography with Orbitrap high-resolution mass spectrometry detection in breakfast cereals

Reviewer 1

The authors report the development of a multi-mycotoxin method for the analysis of several mycotoxins in breakfast cereals and results of the application of this method in a small study with 14 samples from the Italian market.

The method development and validation seems to be scientifically sound in general, but some details still have to be addressed. The results of the market study also need more clarification and discussion.

Taken together the manuscript should not be published in its present form and needs some major revisions before publication.

Specific remarks:

1)Picking up the phrase “multi-mycotoxin method” from the title the authors should explain the selection of mycotoxins they included in their method. Apart from the regulated mycotoxins, the choice of “emerging mycotoxins” seems rather random. Why was fusarenon X (FUS-X) included, but not the structural related nivalenol? Why did the authors omit the well-known deoxynivalenol (DON) derivatives 3- and 15-acetyl-DON and – most important – DON-3-glucoside? All these compounds are commercially available and have been detected in cereals before, so this point needs some clarification.

Authors approached the investigation around the so-called emerging Fusarium toxins, including enniatins A, A1, B and B1 alongside beauvericin, within breakfast cereal samples, in order to provide more evidence about the extensive occurrence of these non-legislated mycotoxins, our choice was guided by the available literature and the prevalence of these mycotoxins in cereal-based products. However, we acknowledge that the selection of mycotoxins may not be exhaustive and further research may be needed to assess the occurrence of other emerging mycotoxins. Although the authors well-Know the importance of deoxynivalenol derivatives doesn't include these mycotoxins in the study because they had no availability to buy reference standards. Sure the authors will conceptualize future investigations including also these mycotoxins, as soon as possible. As we moved forward with the validation study, we obtained good parameters not only for the initially-planned toxins but also for the rest included in the methodology, so authors decided to keep them all for having a more comprehensive study.

2) It is completely astonishing that the authors did not detect deoxynivalenol (DON) in any of their samples. Both literature and legislation are addressing DON as major concern in breakfast cereals, which is contradictory to the results of the authors.

However, the product group “breakfast cereals” is very general and includes different products that have been processed in varying ways that may alter mycotoxin contamination. Authors should give more details about their samples, because this might help to clarify the mycotoxin distribution.

Furthermore, it would have been helpful, if the authors had determined real trueness of the method (not recovery as detailed below) by the analysis of certified reference materials or participation in a ring test. This would have dispelled any concerns, if this non-finding of DON in the samples might be a methodical issue.

Anyway, the authors should address this topic in more detail in the manuscript and discuss potential reasons for their non-finding of certain mycotoxins in their samples.

We appreciate the opportunity to clarify and provide additional information about our findings. With regard the typology of matrix, all the acquired ready-to-eat products were based on (i) only corn (79% of samples); (ii) rice, wheat and barley (14% of samples); (iii) wheat and oat (7% of samples). Three different confections of each product belonging to the same lot were acquired from local markets, homogenized and analyzed as a composite sample, in order to increase the representativity of the sample. Fourteen different typologies of ready-to-eat breakfast cereals were analyzed following this procedure. Authors clarified this in Lines 301-307. Since all the samples were based on wheat or corn, which are susceptible crops to DON accumulation, it was also expected in the final products. In this context, authors, that remarked DON as a major mycotoxin in breakfast cereals in the manuscript, agree with reviewer about the absence of DON in the assayed samples. The analytical methodology validated in our work was capable of quantifying DON at levels above 12.5 µg/kg, sixteen times lower than the requirements set by Regulation (EC) no. 1881/2006 of the Commission for processed cereal-based foods and baby foods for infants and young children (maximum levels of 200 µg/kg), which represent the minimum limit established for the DON in the type of product. However, it is possible that some of the samples tested may be contaminated with DON levels below the limits of the validated method. This could explain the lack of detection in our study, and it highlights the need for continued monitoring and research in this area. We understand the importance of demonstrating the reliability and robustness of our analytical methodology. Nonetheless, the here-applied analytical methodology showed a good performance during the validation study for a reliable quantification of DON in the matrix of study. As suggested, authors addressed the absence of several mycotoxins in Lines 187-195 as follows:

Unlike other studies on breakfast cereals samples, none of the samples tested positive for DON, which is known for its impact in wheat and maize-based products. Mitigation strategies have been extensively discussed over the last years in order to reduce the impact of DON and other relevant mycotoxins in foods and feeds, especially pushed forward by initiatives such as the MycoKey project, that provided practical guides for achieving this reduction on the burden of those mycotoxins [21]. In this line, meta-analysis of DON and other trichothecenes in cereal-based products revealed a decrease over the years regarding concentration and prevalence [10,11].

3) A value named “coefficient of linearity (r2)” (line 122) is not existing. This phrase must not be used. The authors are referring to the “coefficient of determination (R2)”, which is a descriptive value obtained from linear regression. However, the coefficient of determination is a very poor criterion to assess linearity and should not be used anymore nowadays. There are better methods to address this topic like the Lack-of-fit-test or Mandel’s test. Please confer: https://www.researchgate.net/publication/46430384_On_the_misuse_of_the_correlation_coefficient_to_assess_linearity_of_calibration_curves

Authors corrected the wrong terminology within the manuscript. Authors thank reviewer for the provided information that will take into consideration for the next study.

4) Trueness (line 327) cannot be determined by spiking experiments, but only by analyzing certified reference material (CRM). Commission Regulation 2002/657/EC allows that trueness of measurements is assessed through recovery studies, if no CRMs are available. However, there are CRMs available for some mycotoxins in cereals, so this exception from the rule cannot be claimed by authors in general. The validation studies that they have performed were recovery studies and should be named like that.

Authors clarified the fact that trueness was not evaluated, but recovery studies.

5) Table 2 and throughout the whole manuscript: It is common practice in the mycotoxin field to report quantitative values as “µg/kg” and not “ng/g”. Legal limits for mycotoxins in Commission Regulation 1881/2006 are also given in “µg/kg” and the authors are asked to adapt their values accordingly.

Authors changed the units to µg/kg throughout the whole manuscript.

6) The authors should give more details how they determined their LOQ. Were the values obtained from spiking the blank matrix?

Authors spiked blank matrix samples for building a calibration curve. After evaluating the linear range, LOQs were determined as the lowest calibration level inside the linear range, considering a maximum deviation of 20% compared to the theoretical value and a mass error below 5 ppm for each analyte, even if the signal to noise ratio was above 10. This was clarified in Lines 397-400.

7) What kind of breakfast cereal was used as blank matrix for evaluation of the matrix effect?

One of the selected samples included within the study was used for the optimization of sample preparation prior to the validation study. As analysis went further, authors observed the absence of targeted analytes within that product, so this specific sample (based on only corn, as the majority of the selected samples) was also used for the validation study.

8) Figure 1: please check if a) and b) need to be swapped to fit the description in the figure caption. The abbreviations “%R” and “%SSE” should also be defined in the figure caption to facilitate reading.

 The authors thank the reviewer for the precious suggestions.

Reviewer 2 Report

Dear Authors,

 I am pleased to have opportunity to review your manuscript describing the study on multi-mycotoxin method development and occurencen in breakfast cereals. The manuscript is very relevant and informative. It is systematic and well structured, the text is consistent, written in a clear manner and can be read smoothly. However, slight language improvement is recommended. The manuscript is recommended after minor corrections according to the comments bellow.

1.        The title: Regarding the content of the manuscript, I suggest adding the second part of the title (e.g., “Multi-mycotoxin method development using ultra-high liquid chromatography with Orbitrap high-resolution mass spectrometry and mycotoxin occurrence in breakfast cereals in Campania region, Italy”).

2.        Line 60: I suggest adding the short review of available articles as justification why the survey of mycotoxin occurrence in breakfast cereals was necessary (I suggest using the text from lines 171–175 and parts of the text from lines 180–211).

3.        Line 74: I suggest adding “and discussion”.

4.        Line 94: I suggest changing “the lowest volume” to “the lowest volume of acetonitrile”. However, I do not see the statement in line 95 on “excessive dilution” appropriate. To my opinion, the dilution in method C was the largest, because the volume of acetonitrile and water was the largest.

5.        Lines 98–99: I do not think the conclusion that low acidification provides better recoveries can be made. The acidification in method B was the same as in method C, but the recoveries were lower. In method A with higher acidification, there was no water used. Maybe the water/acetonitrile ratio is important? I suggest modifying the text in lines 94–99.

6.        Figure 1, legend: I suggest removing the grey square and the text “120% < %SSE” from the legend below the graphs and mentioning in the text that no SSE and no recovery was above 120%.

7.        Figure 1b: As I understand from the text, 21% of mycotoxins did not reach 70% recovery, so I suppose only 21% of the Method A bar shall be black.

8.        Figure 1: However, for the sake of simpler reading I suggest changing the order of figure 1a (%SSE) and figure 1b (%R), because in the text, first the recovery is reported (lines 86–92) and later SSE (lines 100–109).

9.        Lines 119–120: “Commission Regulation 2002/657/EC” shall be changed to “Commission Decision 2002/657/EC”. However, it was repealed by Commission Implementing Regulation (EU) 2021/808. Further, the performance criteria for some mycotoxins are given in Commission Regulation (EC) No 401/2006. Please, think over how to reference properly.

10.    Line 130: I suggest adding the reference and the relevant maximum levels.

11.    Line 133: I suggest giving a reference where the requirement for inter-day precision below 20% is given.

12.    Table 2: I suggest giving the full names of mycotoxins in the text before the abbreviations appear. They are given systematically in 4.1 Chemicals, Reagents and Materials, but it would be appropriate to give them earlier in the text. Further, as I can see, also AFM1 was included in the validation. Is it reasonable to include it in a method used for cereals? It appears only in milk.

13.    Lines 138–147: I suggest moving the text to the chapter Introduction. 

14.    Line 152: I suggest changing “A 93% of the samples …” to e.g., “Altogether, 93% of the samples …”.

15.    Table 3: Why only nine mycotoxins were included in the table? If other mycotoxins were not detected, I suggest mentioning it in the text (I apologise if I overlooked it), otherwise I suggest adding them to the table.

16.    Lines 171–211: As suggested above, please, think over to move parts of the text to the chapter Introduction and use it here only for comparison/explanation of your results.

17.    Lines 189–191: I think, the incidence and the mean concentrations cannot be compared directly if LOQs are different. You stated that the incidence is constant but occurring at lower concentrations. Maybe Lo Magro et al. did not detect these concentrations because they were below their LOQ. If they could, the incidence would be higher (?) Maybe, therefore also their mean concentrations were almost twice the values reported in your study because the low concentrations were not included in their calculation of mean values. Please, check it and modify the discussion if necessary.

18.    Lines 202–211: I suggest mentioning the conclusions/recommendations from EFSA scientific opinion on beauvericin and enniatins to explain/support the absence of MLs and justify your study. It can be added also into Introduction and/or Conclusions.

19.    Line 249: I suggest changing “diluting 1 mg in 1 mL” to “dissolving 1 mg in 1 mL”.

20.    Chapter 4.3: Please add the description of the Methods A and B.

21.    Lines 317–318: The same comment regarding the legislation as above (Commission Decision 2002/657/EC was repealed by Commission Implementing Regulation (EU) 2021/808, the performance criteria for some mycotoxins are given in Commission Regulation (EC) No 401/2006). Please, think over how to reference properly.

Author Response

Journal Toxins (ISSN 2072-6651)

Manuscript ID: toxins-2199068

Title: Multi-mycotoxin method development using ultra-high liquid chromatography with Orbitrap high-resolution mass spectrometry detection in breakfast cereals

 Reviewer 2

Dear Authors,

I am pleased to have opportunity to review your manuscript describing the study on multi-mycotoxin method development and occurrence in breakfast cereals. The manuscript is very relevant and informative. It is systematic and well structured, the text is consistent, written in a clear manner and can be read smoothly. However, slight language improvement is recommended. The manuscript is recommended after minor corrections according to the comments bellow.

  1. The title: Regarding the content of the manuscript, I suggest adding the second part of the title (e.g., “Multi-mycotoxin method development using ultra-high liquid chromatography with Orbitrap high-resolution mass spectrometry and mycotoxin occurrence in breakfast cereals in Campania region, Italy”).

Authors have introduced the suggested part in the title.

  1. Line 60: I suggest adding the short review of available articles as justification why the survey of mycotoxin occurrence in breakfast cereals was necessary (I suggest using the text from lines 171–175 and parts of the text from lines 180–211).

Authors preferred to maintain those references where placed for avoiding their overuse. Instead, authors justified the monitoring of non-regulated mycotoxins, such as enniatins, using the scientific opinion released by the EFSA CONTAM Panel as follows (Lines 238-248):

Therefore, in this context, the present study provides more evidence about the ubiquity of these toxins that are not regulated and whose toxicology remains under investigation. Despite other surveys have assessed the presence of mycotoxins in breakfast cereals marketed in other countries [19,20,31], ENNs and BEA are not usually included in the analytical methodologies and the focus is mainly put in DON, ZEN, AFs and FBs, which are the regulated toxins. Furthermore, extensive exposure to ENNB and ENNB1 have been observed by previous biomonitoring studies in Italian population [32,33], encouraging the monitoring of these toxins in food products in order to have a more in-depth knowledge about their impact as recommended by the EFSA CONTAM Panel [14].

  1. Line 74: I suggest adding “and discussion”.

Authors added the missing part.

  1. Line 94: I suggest changing “the lowest volume” to “the lowest volume of acetonitrile”. However, I do not see the statement in line 95 on “excessive dilution” appropriate. To my opinion, the dilution in method C was the largest, because the volume of acetonitrile and water was the largest.

As suggested, authors introduced the proposed sentence. Referring to “excessive dilution”, authors meant only the extraction solvent, which is only acetonitrile. In order to clarify this, the sentence in Lines 107-109 was modified to:

“…meaning that probably using too much solvent of extraction may lead to an excessive dilution of the analytes hampering their further determination through LC-HRMS.”

  1. Lines 98–99: I do not think the conclusion that low acidification provides better recoveries can be made. The acidification in method B was the same as in method C, but the recoveries were lower. In method A with higher acidification, there was no water used. Maybe the water/acetonitrile ratio is important? I suggest modifying the text in lines 94–99.

As pointed by the reviewer, there is no enough evidence to make that statement. Authors remove that statement from the text and introduced some more sentences in Lines 121-126 as follows:

“These results showed that this typology of samples contained interferents, forcing to introduce a step to remove them prior to analysis. Method B and Method C used both water as a way minimize the interference of the matrix by diluting. In light of the obtained results, the amount of water used in Method B was not enough for reaching suitable SSE. Method A, instead, used a clean-up step for the removal of contaminants using C18 that also showed to be insufficient to achieve negligible matrix effect.”

  1. Figure 1, legend: I suggest removing the grey square and the text “120% < %SSE” from the legend below the graphs and mentioning in the text that no SSE and no recovery was above 120%.

As suggested, authors removed those part from the legend.

  1. Figure 1b: As I understand from the text, 21% of mycotoxins did not reach 70% recovery, so I suppose only 21% of the Method A bar shall be black.

As pointed by reviewer, the figure was wrong. Authors double-checked the processed data in order to make sure that the data provided was right. A new Figure 1 has been introduced.

  1. Figure 1: However, for the sake of simpler reading I suggest changing the order of figure 1a (%SSE) and figure 1b (%R), because in the text, first the recovery is reported (lines 86–92) and later SSE (lines 100–109).

As suggested by reviewer, authors changed the order to be more logic with the text.

  1. Lines 119–120: “Commission Regulation 2002/657/EC” shall be changed to “Commission Decision 2002/657/EC”. However, it was repealed by Commission Implementing Regulation (EU) 2021/808. Further, the performance criteria for some mycotoxins are given in Commission Regulation (EC) No 401/2006. Please, think over how to reference properly.

As suggested, the reference was changed to “Commission Decision 2002/657/EC”. Authors checked the provided legal texts, but Commission Implementing Regulation (EU) 2021/808 only applies for pharmacologically active substances used in food-producing animals and Commission Regulation (EC) No 401/2006 applies for official analysis. Nevertheless, authors will take these documents into consideration for further studies if needed.

  1. Line 130: I suggest adding the reference and the relevant maximum levels.

As suggested, authors added maximum limits.

  1. Line 133: I suggest giving a reference where the requirement for inter-day precision below 20% is given.

Authors followed the guideline provided  in Commission Decision 2002/657/EC with regards to the CVs of the means in quantitative methods. As suggested, this reference was introduced.

  1. Table 2: I suggest giving the full names of mycotoxins in the text before the abbreviations appear. They are given systematically in 4.1 Chemicals, Reagents and Materials, but it would be appropriate to give them earlier in the text. Further, as I can see, also AFM1 was included in the validation. Is it reasonable to include it in a method used for cereals? It appears only in milk.

Authors removed the names from Section 4.1. and placed them in Lines 86-91 as follows:
“Targeted mycotoxins were: aflatoxins B1, B2, G1, G2 and M1 (AFB1, AFB2, AFG1, AFG2 and M1), beauvericin (BEA), DON, enniatins A, A1, B and B1 (ENNA, ENNA1, ENNB and ENNB1), FB1, FB2, fusarenon X (FUS-X), HT-2 toxin (HT-2), neosolaniol (NEO), ochratoxin A (OTA), T-2 toxin (T-2), α-zearalenol (α-ZEL), α-zearalanol (α-ZAL), β-zearalenol (β-ZEL), β-zearalanol (β-ZAL), zearalanone (ZAN) and ZEN”.

Regarding AFM1, it was included in the LC-HRMS methodology previously developed for dairy products as referenced in the manuscript. As authors moved forward with the validation study, a good performance for AFM1 was obtained so it was kept for further analysis. Nevertheless, as reviewer remarked, there is no point to include AFM1 in analysis of cereal samples so it was removed from the manuscript.

  1. Lines 138–147: I suggest moving the text to the chapter Introduction. 

Authors consider that the text is important to understand the advantage of the here-optimized and validated methodology, which is able to quantify more mycotoxins with similar or even higher sensitivity than the most recently published methodologies for the quantification of mycotoxins in breakfast cereals samples through a simple sample treatment. Nevertheless, authors included a new part in the Introduction section that addresses this in Lines 69-72 as follows:

This is reflected by several recent methodologies for the quantification of mycotoxins in breakfast cereals, that uses different approaches such as enzyme-linked immunoassays (ELISA) procedures or solid-liquid extraction for specific groups of mycotoxins.

  1. Line 152: I suggest changing “A 93% of the samples …” to e.g., “Altogether, 93% of the samples …”.

As suggested, authors rephrase that sentence.

  1. Table 3: Why only nine mycotoxins were included in the table? If other mycotoxins were not detected, I suggest mentioning it in the text (I apologise if I overlooked it), otherwise I suggest adding them to the table.

Authors only reflected the mycotoxins found in the samples in Table 3. To clarify this, authors introduced the following sentence in Line 173:

“Mycotoxins found in the samples and their occurrence data are shown in Table 3.”

  1. Lines 171–211: As suggested above, please, think over to move parts of the text to the chapter Introduction and use it here only for comparison/explanation of your results.

Authors provided studies about mycotoxins in cereal breakfast marketed in Italy in this section. In order to fit the reviewer’s requirement, some parts were restructured, as explained in the next point. After considering it, authors decided to keep this part within the Results and discussion section.

  1. Lines 189–191: I think, the incidence and the mean concentrations cannot be compared directly if LOQs are different. You stated that the incidence is constant but occurring at lower concentrations. Maybe Lo Magro et al. did not detect these concentrations because they were below their LOQ. If they could, the incidence would be higher (?) Maybe, therefore also their mean concentrations were almost twice the values reported in your study because the low concentrations were not included in their calculation of mean values. Please, check it and modify the discussion if necessary.

Authors understand reviewer’s point, the reasoning reflected in the manuscript is indeed misleading. At a quantitative level, lower sensitivity will translate into higher mean concentrations, authors corrected this in the manuscript. On the other hand, the prevalence observed in the present study is similar to the prevalence observed by Lo Magro et al., but with lower sensitivity. This means that, with a higher sensitivity, Lo Magro et al. could have detected either a similar or a higher prevalence compared to our study. Considering that this is also a risky statement, authors changed this part of the Results and discussion section to be as scientifically strict as possible.

  1. Lines 202–211: I suggest mentioning the conclusions/recommendations from EFSA scientific opinion on beauvericin and enniatins to explain/support the absence of MLs and justify your study. It can be added also into Introduction and/or Conclusions.

Authors included the proposed reference in Line 248.

  1. Line 249: I suggest changing “diluting 1 mg in 1 mL” to “dissolving 1 mg in 1 mL”.

As suggested, authors change the phrasing.

  1. Chapter 4.3: Please add the description of the Methods A and B.

As suggested, authors described Methods A and B within Chapter 4.3.

  1. Lines 317–318: The same comment regarding the legislation as above (Commission Decision 2002/657/EC was repealed by Commission Implementing Regulation (EU) 2021/808, the performance criteria for some mycotoxins are given in Commission Regulation (EC) No 401/2006). Please, think over how to reference properly.

As previously indicated, references were checked.

The authors thank the reviewer for the precious suggestions.

Round 2

Reviewer 1 Report

in my opinion the authors have addressed the relevant points in the revised manuscript and their letter, respectively. I have no further objections to publication of the revised manuscript in Toxins.